# Towards capability-adjusted life years in public health and social welfare: Results from a Swedish survey on ranking capabilities

**Anna Månsdotter[1], Björn Ekman[2], Kaspar Walter Meili** [1] *, **Inna Feldman[1,3], Lars Hagberg[4], Anna-Karin Hurtig[1], Lars Lindholm[1]**

1 Department of Epidemiology and Global Health, Umeå University, Umeå, Sweden, 2 Department of Clinical Sciences, Malmö (IKVM), Division of Social Medicine and Global Health (SMGH), Lund University, Lund, Sweden, 3 Department of Public Health and Caring Science, Uppsala University, Uppsala, Sweden, 4 University Health Care Research Center, Faculty of Medicine and Health, Region Örebro University, Örebro, Sweden

* kaspar.meili@umu.se

## Abstract

### Introduction

The aim of this study was to rank capabilities and suggest a relevant set of capabilities for the Swedish context to inform the development of capability-adjusted life years (CALYs). CALYs is a quality of life measure for policy making based on the capability approach by Amartya Sen.

### Materials and methods

A Swedish governmental review proposed the following 10 relevant capabilities: time, financial situation, mental/physical health, political resources, knowledge, living environment, occupation, social relations, security, and housing. Researchers in health-related disciplines from 5 universities ranked these capabilities from 1 to 10 (most to least important) in a web-based cross-sectional survey; 115 of 171 responses were eligible.

### Results

Health, social relations, and financial situation were deemed most important. Stratification by gender, research field, and age group revealed few differences. We found that it was possible to rank capabilities and that health, social relations, and financial situation were ranked highest by a non-representative sample of researchers and doctoral students from health-related disciplines at five Swedish universities.

### Conclusions

The revealed ranking is dependent on the metric and must be further explored. The findings support continued development of CALYs for monitoring and evaluating outcomes in public health and social-welfare interventions.

**Data Availability Statement:** The data underlying this study are available from the Swedish National

Data repository (https://doi.org/10.5878/r2nm-zc35).

**Funding:** The study was funded by two grants from the former Research Council for Health, Working life, and Welfare, now called Forte (Network Grant No 2014-1452 and Grant No 2018-00143, principal investigator Lars Lindholm, https://forte.se/). The founders had no role in study design, data collection and analysis, decision to publish, or preparation of the manuscript.

**Competing interests:** The authors declare that they have no competing interests.

# Introduction

## Background

Monitoring and evaluation within the areas of public health and social welfare involve conceptual and analytical challenges. In theory, cost-benefit analyses based on willingness-to pay (WTP) can be used for evaluating diverse outcomes and even comparisons between sectors, as in principle, all positive and negative outcomes will influence the stated WTP. However, WTP studies are very rare. Instead, in evaluations with a societal perspective, the estimation of productivity gains and changes in resource use are most common. For sectorial perspectives such as health care or social care, often only changes in resource use are included. Sometimes cost-effectiveness analyses or cost-utility analyses based on quality-adjusted life years (QALYs) are used when evaluating intersectoral public health interventions [1, 2].

Yet, the intrinsic value of education or employment [1–3] or non-health benefits of lifestyle behavior change are not properly captured in QALYs [4], and current evaluation approaches may also omit equity aspects [1, 3, 5]. Therefore, we launched capability-adjusted life years (CALYs), a measure that takes into consideration a broad range of capabilities, as outlined in our commentary [6]. In this study, we take one of the initial steps towards developing CALYs.

## Theoretical issues

According to the welfarist framework, welfare (or utility) should be the outcome of interest when assessing the benefits of societal resource allocations [7]. Welfarism is appealing because of the multi-characteristics of public health and social interventions, but the impossibility of making interpersonal comparisons and the difficulty of attaching monetary values to welfare gains continue to hinder its usefulness in applications. Another limitation is that individuals may adapt to the lack of vital necessities and, hence, appear to be quite fine based on self-assessed welfare, when arguably they are not [8].

These limitations led to the development of the framework of extra-welfarism, which is based on the idea that additional information is needed for guiding societal policies [9, 10]. For example, according to Sen, the most important information to consider is *capabilities*, which refer to the opportunities to achieve a flourishing life according to an individual's own wishes [11, 12].

An application of extra-welfarism in health economics are QALYs whereby health care interventions are judged by their impact on health [13, 14]. The QALY concept entails adding up lifetime that is weighted between 0 (death) and 1 (full health), and to use this weighted-lifetime as an outcome measure when calculating cost-effectiveness (cost per QALY) of health care and public health interventions [14]. Prominent examples include the different EQ-5D questionnaires that measure health-related outcomes in 5 attributes and 3 to 5 levels, which form combinations of health profiles with associated quality weights. The quality weights for EQ5D measures are derived from information about trade-offs between the different health profiles, usually gathered in setting-specific valuation studies [14].

The focus on health as opposed to utility and allowing interpersonal comparisons are defining extra-welfarist properties of QALYs [15]. Due to the extra-welfarist nature, the QALY concept is in theory not restricted to measuring health benefits but in practice is used mostly to evaluate health consequences. Cookson [16] suggested a 'capability QALY' interpretation of QALYs as a 'cardinal index of an individual's capability set'. We see CALYs, as outlined below, as a specific implementation according to Cookson's interpretation.

## Summary measure for policy

Since public health interventions may impact other well-being components besides health and since social welfare policy and reform (education, labour market, social insurance, etc.) may also affect lifetime health, it seems meaningful to establish a summary measure of capabilities. This requires the acceptance that one capability (e.g., income) can be traded for another capability (e.g., health) [17] and that agreement can be reached on the relative values of the included capabilities [12]. For a specific context, it is possible to distinguish different kinds of capabilities based on the extent to which a society ought to make them available to an individual. In democratic high-income countries such as Sweden, one category comprises capabilities that everybody has an undisputed right to have access to, such as political resources or a minimal threshold of education through basic schooling. For other capabilities, there may be less consensus that everybody should have them to the same extent, for example, the capability to be wealthy enough to buy property or to have an occupation that one is always completely happy with. Hence, important questions to ask are the following: (a) Which capabilities are sensitive to public polices? (b) Which capabilities can best explain differences in quality of life? An example relevant to the Swedish context may be that the capabilities of beauty and love are important but not easy targets for public decision making, whereas access to clean water and air does not discriminate sufficiently since all inhabitants have these capabilities to a reasonable extent.

Another acknowledgment is that an applied conceptualisation of the capability approach should take opportunity into consideration as well as achievement. According to Sen, capability is the freedom to achieve valuable functions, which is often illustrated by the difference between voluntary fasting and involuntary starving [18]. A corresponding example from the context of a high-income country could be freely chosen retirement versus involuntary unemployment at the age of 60 years. Nussbaum's point here is that policy should not focus on what people choose to do but their capability to function well if they choose to do something [19]. In empirical analysis, the assumption is often that the functions an individual possesses provide fair understanding of the capabilities that he/she has had access to [20], i.e., a rational person chooses the most attractive functions available in his or her set of capabilities.

## Other approaches and our proposal

Based on the explicit assumption that QALYs may be too restricted for health and social care, Al-Janabi et al. [21] developed a descriptive capability-based system labelled ICECAP-A (ICE-pop CAPability measure for Adults), together with weights linked to key capabilities [22]. According to Mitchell et al. [23], an additional approach in economic evaluations could be to consider shortfalls from 'sufficient capability'. Another example for operationalisation of the capability approach for public health is the OCAP-18 measure by Lorgelly et al. [24].

Our contribution, CALYs, is an outcome measure, for monitoring trends and distributions, as well as evaluating public health and social interventions. CALYs will be calculated based on capability weights for different quality-of-life states, from 0 (worst) to 1 (best). For example, a state that represents the quality of life for a given capability set may have a weight of 0.7 attached, resulting in 0.7 CALYs for a person-year, i.e., the same concept as in QALYs. Whereas QALYs are appropriate in situations where the main interest are health benefits, CALYs aim to be used in situations where the goal is to measure changes in individuals' general capabilities.

Key aspects of CALYs are that we rely on a deductive, rather paternalistic approach, by mainly trusting so-called 'fair-minded people' to select relevant capabilities [25]; that we consider a trade-off between equity and efficiency according to the *fair innings* argument by

Williams (i.e., considering good before perfect years) [26]; and that we rely on revealed public preferences for establishing the threshold value for a CALY [27]. We envision a simple structure for CALY states with a limited number of capability dimensions and levels of achievement, similar to the EQ-5D-3L construction [28], for the purpose of facilitating both the further development and application of CALYs. The definite configurations of dimensions and levels still needs to be determined.

Collapsing different capability levels into a single index may conflict with Sen's suggested capability approach that rejects complete orderings [12]. However, the QALY concept has demonstrated the usability of a simple quality weight for policy making where the advantages of this abstraction enabled efficient applications of extra-welfarist concepts. The preference-based process of eliciting QALY weights also is a pragmatic democratic approach to average the value judgement of representative population samples.

The development of the CALY measure necessitates to develop a procedure to decide which capability dimensions should be included. To our knowledge, no previous work specific to the Swedish context exists that has attempted to assess a relevant ranking of capabilities based on survey data.

In this study, the aim was to examine the possibility of ranking capabilities by their importance for quality of life and to tentatively indicate a selection of most relevant capabilities based on various metrics among Swedish respondents. In relation to the overall project, the purpose was to collect information for the selection of dimensions for the CALY measure. We found that ranking capabilities is feasible and that health, social relations, and financial situation were ranked highest by a non-representative sample of researchers and doctoral students in health-related disciplines at five Swedish universities.

## Materials and methods

### List of suggested capabilities

The Swedish government commissioned an investigation [29] to review, analyse, and propose measures for quality of life. The appointed investigator and experts conducted a literature review. The investigation proposed a list of 10 freedoms of action and achievements to be particularly important for people's quality of life, based on previous lists by Johansson [30], the Report by the Commission on the Measurement of Economic Performance and Social Progress by Stiglitz, Sen, and Fitoussi [31], and the OECD [32]. We considered these as representing a relevant list of capabilities for our purpose, and we added a short interpretation for each in relation to what it means to live well in present-day Sweden, based on their description in the government's investigation [29]:

- Time: experiencing a balance between necessary activities (work, other duties) and voluntary activities (social contacts, physical activity, entertainment, hobbies, etc.).

- Financial situation: having monetary means (salary, other income, savings) that permit an enjoyable living standard.

- Health: having a health status (mental and physical) that does not limit the possibility of working or engaging in other wanted activities.

- Political resources: having the opportunity to form an opinion on minor and major issues and being able to present them while being treated with respect (polling stations, associations, authorities, media, etc.).

- Knowledge: having the education and experience required to work with or engage in interesting issues and purposeful activities.

- Living environment: being in an enjoyable and practical context (neighbours, buildings, parks/nature, transportation, stores, etc.).

- Occupation: having satisfying work or other engagements (studies, internships, household work, care of relatives, etc.).

- Social relations: having access to close relationships (family, friends, acquaintances) that contribute to pleasure and development and to persons that offer advice and support when needed.

- Security: not feeling afraid of being afflicted by burglary, vandalism, or various kinds of threats of violence, either at home or in other places.

- Housing: having affordable and stable housing that is appreciated in terms of functionality, appearance, and location.

## Survey respondents

Potential survey respondents were a convenience sample of Swedish-speaking researchers and doctoral students in the disciplines of public health, global health, epidemiology, medicine, women´s and children´s health, and health care at five Swedish universities. In agreement with Sen's proposal for a context-sensitive adaptation of the capability approach, the survey language was Swedish.

The number of potential respondents was about 830 in total, of which, approximately 100 were from the Department of Public Health Sciences, Karolinska Institutet; 440 were from the Medical Faculty at Lund University; 40 were from the Unit of Epidemiology and Global Health at Umeå University; 180 were from the Department of Women's and Children's Health and the Department of Public Health and Care Science at Uppsala University; and 70 were from the University Health Care Research Center at Region Örebro County. The number of respondents was 171, although 4 of them did not explicitly indicate consent to participate. Hence, the sample size was 167, making up an estimated response rate of 20% (167/830).

## Procedure and questionnaire

An application for complete ethical review was sent to the Stockholm Board for Ethical Clearance (EPN). An advisory statement concluded that EPN had no ethical objections to the study (Protocol 2016/4). Informed consent to participate was obtained from all individual respondents included in the study, and the study was conducted according to the principles expressed in the Declaration of Helsinki.

The invitation to participate in the web-based survey was sent by email (28 May, 2016), with information about the aim, invited participants, ethical statement, contact information, and deadline (15 June, 2016). The survey was accessible at once. The attached documents consisted of an informational letter (background, aim, task, estimated time taken, risks and benefits, handling of data, presentation of results, and voluntariness) and a summarised research plan (the overall project and the study). This was followed by a reminder (10 June, 2016).

First, the respondents were asked to give informed consent to participate in the survey by ticking a checkbox. The respondents then ranked the list of 10 capabilities from 1 (the most important capability) to 10 (the least important capability). The instructions were that any rank (1, 2, 3, . . ., 10) could be used only once to arrive at a unique ordering. The intent of using a ranking task was to promote trade-offs between the capabilities and avoid participants indicating all capabilities as important. Finally, the respondents were asked to enter gender

(male, female, other, empty), age (<30 years, 30–39 years, 40–49 years, 50–59 years, 60+ years, empty), research field (free text), and potential comments (free text). The original Swedish and translated English questions from the survey are provided in S1 Questionnaire. For later analysis, we categorized the free-text research field answers into medicine, public health, social science, and other. We pre-tested the survey informally in our networks outside academia.

Sen (1998) has argued that health is probably the best indicator of people's well-being if one has to select only one, and the respondents were researchers in health-related disciplines. This led us to assume that some respondents may have fulfilled the task correctly but with an inverse ranking from 10 (the most important capability) to 1 (the least important capability) if they assigned health, together with social relations or financial situation, a rank greater than 5. To assess the impact of potentially inverse rankings, a sensitivity analysis was conducted without responses where health was assigned a greater than 5.

## Statistical analysis

We calculated means and medians overall and stratified by gender, research field, and age group. We also calculated the cumulative number of times a capability was ranked 1 to 2, 3, 4, or 5. We used bootstrapping with 10,000 resamples to calculate confidence intervals for means and medians because the underlying rank data may not have been normally distributed. Furthermore, we dissected the rankings into pairwise comparisons of two capabilities and examined for each pair the number of times one capability was ranked ahead of the respective other, hereafter termed the *win count balance*. For example, a win count balance of 5 for A over B means that A was ranked 5 times ahead of B. We used the win count balance to look for systematic patterns among the combinations. We used R 3.6 for all analyses [33].

## Results

Among the 167 respondents who consented, 112 respondents (67%) completed the task correctly (i.e., assigned 10 unique ranks to the 10 capabilities), whereas 55 respondents (33%) completed the task incorrectly (i.e., incomplete rankings). Three incorrect responses (2%) contained 9 unique ranks, which made it possible to impute the missing rank. The main results presented below are based on the correct responses plus the 3 imputed responses (n = 115).

Study participants were mostly female (74 women compared with 39 men and 2 unanswered) and came from health-related fields (29 from medicine and 40 from public health compared with 21 from social science, 9 other 16 unanswered). All participants except for 3 were older than 30 years old. The lowest age category was excluded from stratified analysis because of the low strata size. **Table 1** shows the characteristics of participants.

**Fig 1** shows the distribution of ranks per capability dimension. Health was most often ranked first, social relations was most often ranked second, and financial situation was most often ranked fifth. Distributions were less accentuated and more broadly distributed for knowledge, security, time, political resources, occupation, housing, and environment. Political resources, housing, and environment had 10, 9, and 8 as the most frequently assigned ranks.

Each cell in **Table 2** shows the number of times the capability in the row was ranked ahead of (positive values) or behind (negative values) than the capability in the column (win count balance). We discovered a pattern where the binary relation that consists of all combinations of capabilities with positive win count balances forms a strict total order for a set of five capabilities [34]: negative row values before and positive row values after the 0 diagonal values imply health > social relations > financial situation> knowledge > occupation, although sometimes by very small margins (e.g., occupation vs. time and knowledge vs. security). In other words, for the five capabilities, for any capability A and another capability B, if A was

**Table 1. Participant's characteristics.**

| Characteristic | Category | N | Proportion |
|---|---|---:|---|
| **Gender** | Woman | 74 | 0.64 |
| | Man | 39 | 0.34 |
| | NA | 2 | 0.02 |
| **Field** | Medicine | 29 | 0.25 |
| | Public Health | 40 | 0.35 |
| | Social science | 21 | 0.18 |
| | Other | 9 | 0.08 |
| | NA | 16 | 0.14 |
| **Age group** | 30 and lower | 3 | 0.03 |
| | 30–39 | 43 | 0.37 |
| | 40–49 | 26 | 0.23 |
| | 50–59 | 24 | 0.21 |
| | 60 and over | 19 | 0.17 |
| **Overall** | | 115 | |

ranked ahead of B more often and if B was ranked ahead of a third capability C more often, then A was also ranked ahead of C more often. The other five capabilities (time, security, political resources, housing, living environment) were dominated by the above five capabilities, but they did not reveal similar patterns.

Table 2 also indicates the cumulative count of a capability having been ranked 1 to 5. By this metric, health (87) was deemed most important, followed by social relations (75) and financial situation (67), whereas living environment was ranked least important (40). The counts for knowledge, security, time, political resources, occupation, and housing were clustered together (48 to 56).

Table 3 depicts the overall median ranks by gender (women and men), research field (public health, medicine, social sciences, and other), and age group. Overall, the capability of health was top-ranked (median rank 2), followed by social relations (4) and financial situation (5), whereas housing (7) and living environment (7) were bottom-ranked. The 95% confidence intervals (CIs) for the medians of health and social relations did not overlap, whereas the CIs for the other medians all included rank 5, with an exception for living environment, with a 95% CI of 6–7.

When stratifying by gender, research field, and age group, health and social relations had the top 2 median ranks, except for the research field 'other', where social relations was positioned behind financial situation and occupation. Housing and living environment consistently had a median rank of 6 to 8 in the different strata, apart from rank 5 for housing for the age group 30–39 years. Median ranks across research fields were mostly similar. Across age groups, the mean rank of social relations decreased with increasing age, from 5 for 30–39 years to 2 for 60+ years. The median rank of time for age groups 30–39 years and 40–49 years contrasted with the median rank of age groups 50–59 years and 60+ years (5 and 4 vs. 7.5 and 7, respectively).

Table 4 depicts the bootstrapped overall mean ranks for the same strata as for the median ranks (Table 3). Health and social relations were ranked highest on average (3.43 and 4.48, respectively), and housing and living environment were ranked lowest (6.21 and 6.35, respectively). Mean ranks differed relatively little for the 8 bottom-ranked capabilities and were clustered between 5.37 and 6.35. Health and social relations were consistently among the top 3 mean ranks across all strata. The mean rank of financial situation was the third highest for

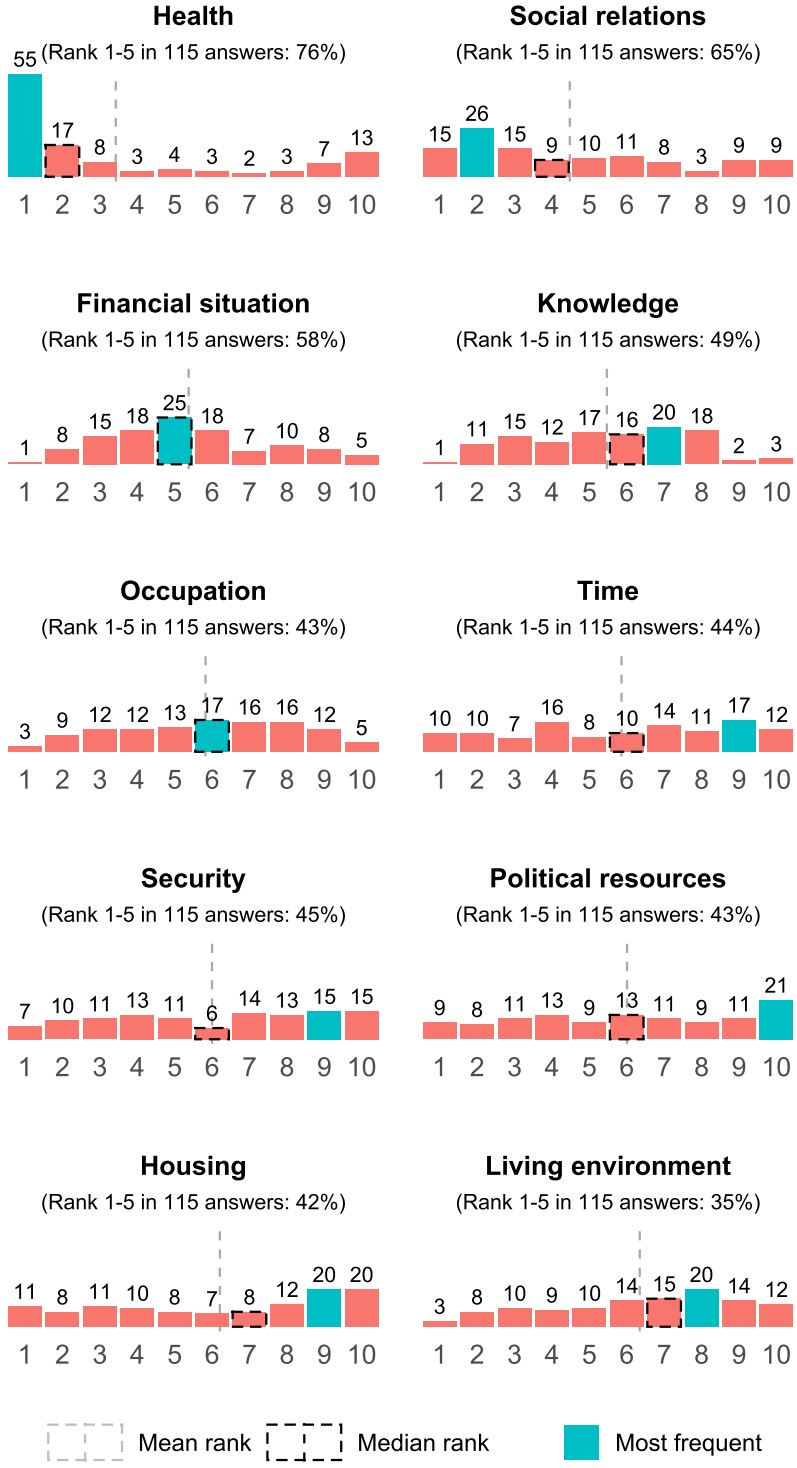

**Fig 1. Distribution of ranks per dimension.** Distribution of ranks for all capability dimensions, with the number of rankings above the bars. Teal indicates the most frequently chosen rank, the dashed grey line is the mean rank, and the dashed black rectangle is the median rank. Subheadings denote the proportion of times a dimension was ranked 1–5 out of the 115 answers.

**Table 2. Win count balance for each pair of capabilities.**

| Ranking of importance | Win count balance (of the total 115 pairwise comparisons) | | | | | | | | | | Ranked 1 to 5 |
|---|---|---|---|---|---|---|---|---|---|---|---|
| | Health | Social relations | Financial situation | Knowledge | Occupation | Time | Security | Political resources | Housing | Living environment | |
| Health | 0 | 39 | 51 | 61 | 61 | 49 | 59 | 47 | 55 | 55 | 87 |
| Social relations | −39 | 0 | 23 | 27 | 41 | 27 | 31 | 37 | 35 | 53 | 75 |
| Financial situation | −51 | -23 | 0 | 5 | 13 | 15 | 13 | 5 | 25 | 27 | 67 |
| Knowledge | −61 | −27 | −5 | 0 | 23 | 11 | 3 | 23 | 17 | 23 | 56 |
| Occupation | −61 | −41 | −13 | −23 | 0 | 1 | 17 | 5 | 29 | 11 | 49 |
| Time | −49 | −27 | −15 | −11 | −1 | 0 | 1 | −3 | 3 | 19 | 51 |
| Security | −59 | −31 | −13 | −3 | −17 | −1 | 0 | 9 | −5 | 5 | 52 |
| Political resources | −47 | −37 | −5 | −23 | −5 | 3 | −9 | 0 | −3 | 9 | 50 |
| Housing | −55 | −35 | −25 | −17 | −29 | −3 | 5 | 3 | 0 | −7 | 48 |
| Living environment | −55 | −53 | −27 | −23 | −11 | −19 | −5 | −9 | 7 | 0 | 40 |

The left section is the number of times (*win count balance*) the capability in a row was ranked ahead of (positive values) or behind (negative values) the capability in the column for each pair of capabilities given by row and column. The right section is the cumulative number of times a capability was ranked 1 to 5.

**Table 3. Median ranks.**

| | Overall | Gender | | Research field | | | | Age group | | | |
|---|---|---|---|---|---|---|---|---|---|---|---|
| Capability | Overall | Men | Women | Medicine | Public health | Social science | Other | 30–39 | 40–49 | 50–59 | 60+ |
| n | 115 | 39 | 74 | 29 | 40 | 21 | 9 | 43 | 26 | 24 | 19 |
| Health | 2 | 1 | 2 | 1 | 2 | 3 | 1 | 2 | 2 | 1.5 | 1 |
| | (CI 1–2) | (CI 1–2) | (CI 1–3) | (CI 1–2) | (CI 1–3) | (CI 1–6) | (CI 1–5) | (CI 1–3) | (CI 1–2.5) | (CI 1–5) | (CI 1–2) |
| Social relations | 4 | 4 | 3 | 3 | 3 | 4 | 6 | 5 | 3 | 2.5 | 2 |
| | (CI 3–5) | (CI 3–5) | (CI 3–5) | (CI 2–6) | (CI 2–5) | (CI 2–7) | (CI 2–7) | (CI 4–6) | (CI 2–5) | (CI 2–6) | (CI 2–5) |
| Financial situation | 5 | 5 | 5 | 5 | 6 | 5 | 4 | 5 | 5.5 | 5 | 4 |
| | (CI 5–6) | (CI 4–5) | (CI 5–6) | (CI 4–6) | (CI 5–6) | (CI 4–7) | (CI 3–8) | (CI 5–6) | (CI 4.5–6.5) | (CI 4–5) | (CI 4–6) |
| Knowledge | 6 | 6 | 5.5 | 6 | 5 | 6 | 6 | 6 | 5.5 | 5 | 5 |
| | (CI 5–6) | (CI 5–7) | (CI 5–6) | (CI 4–6) | (CI 4–7) | (CI 5–7) | (CI 4–8) | (CI 5–6) | (CI 3.5–7) | (CI 4–7) | (CI 3–7) |
| Occupation | 6 | 6 | 6 | 5 | 7 | 6 | 4 | 7 | 5.5 | 6 | 6 |
| | (CI 5–7) | (CI 5–7) | (CI 5–7) | (CI 4–7) | (CI 6–7) | (CI 5–8) | (CI 2–9) | (CI 5–8) | (CI 4–7) | (CI 4–7) | (CI 4–7) |
| Time | 6 | 7 | 5.5 | 7 | 6 | 6 | 8 | 5 | 4 | 7.5 | 7 |
| | (CI 5–7) | (CI 6–8) | (CI 4–7) | (CI 4–8) | (CI 4.5–8) | (CI 3–7) | (CI 3–9) | (CI 4–7) | (CI 3.5–7) | (CI 5.5–9) | (CI 6–9) |
| Security | 6 | 7 | 6 | 6 | 6 | 6 | 6 | 6 | 7.5 | 6.5 | 7 |
| | (CI 5–7) | (CI 4–8) | (CI 5–7) | (CI 4–8) | (CI 4–8) | (CI 4–8) | (CI 4–9) | (CI 4–7) | (CI 4–8.5) | (CI 4–8) | (CI 4–9) |
| Political resources | 6 | 6 | 6 | 6 | 5 | 6 | 8 | 6 | 7 | 6 | 6 |
| | (CI 5–7) | (CI 4–7) | (CI 5–7) | (CI 6–9) | (CI 4–7) | (CI 3–9) | (CI 3–10) | (CI 4–8) | (CI 6–9) | (CI 4–7) | (CI 4–8) |
| Housing | 7 | 8 | 6 | 7 | 8 | 8 | 5 | 5 | 8.5 | 6.5 | 8 |
| | (CI 5–8) | (CI 5–9) | (CI 5–8) | (CI 5–8) | (CI 4.5–9) | (CI 3–9) | (CI 2–6) | (CI 4–8) | (CI 6–9) | (CI 5–8) | (CI 5–9) |
| Living environment | 7 | 7 | 7 | 8 | 7 | 6 | 8 | 6 | 7 | 6.5 | 8 |
| | (CI 6–7) | (CI 6–8) | (CI 6–7.5) | (CI 6–8) | (CI 5.5–8) | (CI 4–7) | (CI 3–10) | (CI 5–7) | (CI 5.5–8) | (CI 5–8) | (CI 7–9) |

Per strata, 95% confidence intervals (CIs) were calculated with bootstrapping based on 10,000 resamples. The number of rankings per subgroup is denoted by n. In the case of an even number of observations, .5 fractions could occur because the median was calculated by averaging the observations positioned at n/2 and n/2 +1.

**Table 4. Mean ranks.**

| Capability | Overall | Gender | | Research field | | | | Age group | | | |
|---|---|---|---|---|---|---|---|---|---|---|---|
| | Overall | Men | Women | Medicine | Public health | Social science | Other | 30–39 | 40–49 | 50–59 | 60+ |
| **n** | 115 | 39 | 74 | 29 | 40 | 21 | 9 | 43 | 26 | 24 | 19 |
| **Health** | 3.43 | 2.64 | 3.8 | 2.76 | 3.38 | 3.86 | 2.56 | 3.84 | 2.69 | 3.75 | 2.63 |
| | (CI 2.8–4.1) | (CI 1.8–3.6) | (CI 3–4.6) | (CI 1.8–3.9) | (CI 2.4–4.4) | (CI 2.5–5.2) | (CI 1–4.6) | (CI 2.8–4.9) | (CI 1.9–3.7) | (CI 2.4–5.2) | (CI 1.5–4.1) |
| **Social relations** | 4.48 | 4.54 | 4.47 | 4.14 | 3.98 | 5 | 5.22 | 5.21 | 4.08 | 4.38 | 3.42 |
| | (CI 3.9–5) | (CI 3.7–5.4) | (CI 3.8–5.2) | (CI 3.1–5.3) | (CI 3.2–4.8) | (CI 3.8–6.3) | (CI 3.6–6.9) | (CI 4.3–6.1) | (CI 3.1–5.1) | (CI 3.1–5.7) | (CI 2.5–4.4) |
| **Financial situation** | 5.37 | 4.74 | 5.64 | 5.17 | 5.65 | 5.57 | 4.78 | 5.63 | 5.62 | 5.12 | 4.74 |
| | (CI 5–5.8) | (CI 4.1–5.4) | (CI 5.1–6.1) | (CI 4.6–5.8) | (CI 5–6.3) | (CI 4.6–6.7) | (CI 3.3–6.2) | (CI 5–6.3) | (CI 4.7–6.5) | (CI 4.4–5.8) | (CI 3.8–5.7) |
| **Knowledge** | 5.47 | 5.59 | 5.39 | 5.38 | 5.35 | 5.81 | 5.89 | 5.72 | 5.31 | 5.46 | 4.84 |
| | (CI 5.1–5.9) | (CI 4.9–6.2) | (CI 4.9–5.9) | (CI 4.7–6.1) | (CI 4.7–6) | (CI 4.9–6.7) | (CI 4.8–7) | (CI 5.1–6.3) | (CI 4.5–6.2) | (CI 4.5–6.4) | (CI 3.9–5.8) |
| **Occupation** | 5.83 | 6 | 5.69 | 5.28 | 6.47 | 6.1 | 4.78 | 6.23 | 5.58 | 5.58 | 5.42 |
| | (CI 5.4–6.3) | (CI 5.2–6.7) | (CI 5.1–6.2) | (CI 4.4–6.2) | (CI 5.9–7.1) | (CI 5.1–7) | (CI 3–6.7) | (CI 5.5–7) | (CI 4.7–6.5) | (CI 4.7–6.5) | (CI 4.5–6.4) |
| **Time** | 5.86 | 6.33 | 5.58 | 6.17 | 5.95 | 5.1 | 6.78 | 5.4 | 5.12 | 6.67 | 7.16 |
| | (CI 5.3–6.4) | (CI 5.5–7.2) | (CI 4.9–6.2) | (CI 5.1–7.2) | (CI 5.1–6.8) | (CI 3.9–6.4) | (CI 5–8.3) | (CI 4.5–6.3) | (CI 4.1–6.2) | (CI 5.4–7.8) | (CI 6.2–8.1) |
| **Security** | 6 | 6.15 | 5.96 | 5.97 | 5.83 | 5.67 | 6.56 | 5.67 | 6.31 | 6.04 | 6.21 |
| | (CI 5.5–6.5) | (CI 5.3–7.1) | (CI 5.3–6.6) | (CI 5–6.9) | (CI 4.8–6.8) | (CI 4.6–6.7) | (CI 5.1–8) | (CI 4.9–6.5) | (CI 5.1–7.5) | (CI 4.9–7.1) | (CI 4.9–7.5) |
| **Political resources** | 6.01 | 6.03 | 6 | 6.83 | 5.42 | 5.76 | 7.11 | 5.86 | 6.73 | 5.58 | 6.11 |
| | (CI 5.5–6.5) | (CI 5.1–6.9) | (CI 5.3–6.7) | (CI 5.8–7.8) | (CI 4.6–6.2) | (CI 4.3–7.2) | (CI 5.2–8.8) | (CI 4.9–6.8) | (CI 5.7–7.7) | (CI 4.4–6.8) | (CI 4.9–7.3) |
| **Housing** | 6.21 | 6.44 | 6.2 | 6.52 | 6.58 | 6.33 | 4.67 | 5.6 | 7 | 6.25 | 6.84 |
| | (CI 5.6–6.8) | (CI 5.4–7.4) | (CI 5.5–6.9) | (CI 5.5–7.4) | (CI 5.5–7.5) | (CI 4.9–7.7) | (CI 3.2–6.2) | (CI 4.7–6.6) | (CI 5.8–8.1) | (CI 5.1–7.3) | (CI 5.4–8.2) |
| **Living environment** | 6.35 | 6.54 | 6.27 | 6.79 | 6.4 | 5.81 | 6.67 | 5.84 | 6.58 | 6.17 | 7.63 |
| | (CI 5.9–6.8) | (CI 5.7–7.3) | (CI 5.7–6.8) | (CI 5.9–7.7) | (CI 5.6–7.2) | (CI 4.6–7) | (CI 4.7–8.4) | (CI 5–6.6) | (CI 5.6–7.5) | (CI 5.2–7.1) | (CI 6.7–8.5) |

Per strata, 95% CIs were calculated with bootstrapping based on 10,000 resamples. The number of rankings per subgroup is denoted by n.

men, whereas the third-highest mean rank for women was knowledge. The third-highest mean rank differed considerably across age groups and research field; for example, time was the third highest for age groups 30–39 years and 40–49 years, but it was the second lowest for the age group of 60+.

## Sensitivity analysis

Twenty-eight respondents ranked health greater than 5. Of these 28 respondents, 25 also ranked social relations or financial situation greater than 5, together with health. Because health, social relations and financial situation where otherwise ranked 1 to 3 by all metrics; this may indicate reverse rankings. For the sensitivity analysis, in which these 28 potential reverse rankings according to the health criterion were excluded from the 115 respondents in the main analysis, the rankings of importance varied little. The number of times a capability was ranked 1 to 5 resulted in the same first 5 positions (health, social relations, financial situation,

knowledge, security). Similarly, the total order relation on the win count balance with health > social relations > financial situation > knowledge > occupation did not change. The order implied by the mean ranks also largely remained the same; only time and security (position 6 and 7 with all data) switched. The order implied by the median ranks did not change for the top 3 positions, but time, political resources, and security had a median rank of 7 and thus position 5, leaving knowledge and occupation alone at position 4 with a median rank of 6. The most frequent rank only changed for political resources to position 10.

## Discussion

### Main findings

This study represents a step towards CALYs, a novel outcome measure for public health and social welfare. Most of the respondents who consented managed to rank the proposed set of capabilities. This indicates that it is possible to discriminate between these capabilities in terms of their importance for quality of life. The results suggest that participants deemed health to be most important, followed by social relations and financial situation. Knowledge, occupation, time, security, political resources, housing, and living environment were ranked lower, and the exact order depends on the metric used to synthesise the individual rankings.

### Endorsing the capabilities

Our gross list of capabilities consisted of 10 indicators of quality of life, considered to be theoretically sound, empirically valid, policy relevant, and measurable by survey or registry data [29]. To test the validity in terms of representing capabilities, one may pragmatically compare the utilised indicators with the 10 basic capabilities proposed by Nussbaum [19]. For instance, 'Bodily Health' and 'Bodily Integrity' connect to health and security; 'Senses, Imagination, and Thought' connects to knowledge and political resources; 'Other species' connects to the living environment; 'Play' connects to time, social relations, and occupation; and 'Control over One's Environment' connects to housing and financial situation. All in all, we propose that the capabilities considered be endorsed in the sense of being covered by Nussbaum's classical list.

### Ranking of importance and comparison with other studies

Constructing a ranking of importance is challenging. Overall, health is clearly placed first in terms of frequency, mean, median, and win count balance metrics, followed by social relations and financial situation. Differences between positions from 3 to 10 were less pronounced. Placing occupation and knowledge as two of the 5 most important capabilities was supported by the total order relation formed by the win count balance, the mean, the median, and the most frequent metrics. The number of times a capability was ranked 1 to 5, however, did not result in occupation being included in the top five but instead favouring security. The mean, median, most frequent rank, or number of times being ranked from 1 to 5 offered little insight into the fifth most important capability due to the small differences from the subsequent positions. Occupation, time, security, political resources, and housing (but not living environment) are all contenders.

Beyond the limited size and biased nature of the sample, none of the different metrics used here offer a definite answer to the question of importance ranking; they all have individual advantages and disadvantages. The median is appropriate for ordinal data and takes into account the whole distribution of ranks; the most frequent rank reflects the opinion of the largest group, and the total order based on the win count balance relies only on comparisons of two alternatives. However, all of these lack discriminative power or they can be ambiguous.

The mean rank may be less suited to rank data due to its ordinal nature, but it offers more discriminative power since mean ranks are likely to differ, even if the underlying distributions are similar. Counting the rankings between 1 and 5 offers better discrimination, but it only reflects whether a capability ranks in the top 5, and it fails to take into account further distributional differences in each of the two halves of the ranking scale.

A high proportion of participants stated in the free-text reflections that they perceived the ranking task as difficult. Assuming higher variance due to these difficulties, these comments support the result of the empirical analysis, in which only health, social relations, and, to some extent, financial situation were clearly positioned ahead of the other capabilities. Although this qualitative aspect lies beyond our quantitatively focused analysis, such input is valuable for informing the selection process and for refining the survey design to facilitate the ranking as much as possible.

The ranking may be compared with studies by Anand et al. [35] and Al-Janabi et al. [21]. Anand et al. tested the effects of basic capabilities (28 indicators) on well-being (satisfaction with life overall, health, home, income, partner, job, social life, leisure, etc.), controlled for partner and job (having, not having) and personality traits ('always look on the bright side'). The challenge of selecting capabilities, and not simply functionings, was guided by the following two types of questions: first, those that ask about general capabilities, such as health and education, which have implications for what we can do in other areas of life; and, second, those that ask about particular capabilities, such as 'Would you like to be able to pay for a week's annual holiday away from home, but must do without because you cannot afford it?' The authors found a strong argument for considering the capabilities of good health and sexual freedom, social interaction and self-worth, and happiness and amusement. When developing the ICECAP measure, Al-Janabi et al. [21] found particular support for the over-arching capability attributes of feeling settled and secure; love, friendship, and support; being independent; achievement and progress; and enjoyment and pleasure. The supplementary population tariff shows that 'attachment' and 'stability' accounted for around 22% of the total value, whereas 'autonomy,' 'achievement,' and 'enjoyment' accounted for about 18% [22].

Hence, even though our study has limitations, its findings have broader support. To endorse the high rankings of financial situation and knowledge, which were not highly valued in the studies by Anand et al. [35] and Al-Janabi et al. [21, 22], one may refer to the well-known ecological interpretation of the capability approach, i.e., the human development index (HDI), considering gross national product, life expectancy, and literacy [36].

## Limitations

This study was not designed to assess a definite representative ranking order of important capabilities in the Swedish context and should not be used as such. The study participants were not representative of the Swedish population and consisted of researchers in health-related disciplines. Hence, they are likely biased towards valuing health, but also other capabilities such as knowledge and social relations as determinants of health, as particularly important.

Deciding on how many capabilities should be included in the CALY measure based on ranking data may be arbitrary because the ranking task does not directly yield information about the relative strengths of the different capabilities. Additionally, there are limitations regarding the aim of examining the possibility of ranking capabilities per se. First, we do not know whether the low response rate was due to explicitly refusing to rank capabilities or to reasons such as lack of time and interest. Second, we do not know the characteristics of the respondents (researchers and doctoral students) in terms of their ranking capacity. Third, the

web-based survey had technical limitations that need revision, such as allowing for ranking two capabilities with the same number. Similarly, the occurrence of suspected inverse rankings is an indicator that the instructions and visual design of the survey need improvement. Fourth, we did not a priori calculate a required sample size or validate the questionnaire.

## Future research

The limitations of the study confirm our intention to select capabilities based on an approach that, in extra-welfarist fashion, does not only empirically consider individual preferences, namely inviting 'fair-minded people' [25] to discuss the most relevant capabilities for the Swedish context in a participatory Delphi process [37]. We have also explored the calculation of CALY weights in a web-based survey among 2,000 Swedish residents [38]. The study explored discrete choice and time trade-off questions [39] to calculate weights, and anchored them on a 0 (worst) to 1 (best) scale. Among health, social relations, financial situation, knowledge and occupation, the highest-valued capability dimensions in this study were health and social relations. These survey results, together with the results presented here, were used to inform the Delphi process (manuscript in preparation). After the selection of capabilities is finished, we will continue the development of the CALY measure by finalizing the instrument, performing a valuation study for eliciting CALY weights, and finally applying the measure in economic evaluations.

## Conclusions

We conclude that, given clear instructions and alternatives, it is possible to generate a ranking of capabilities by their importance to quality of life. The analysis of the ranking responses shows that health, social relations, financial situation, and potentially knowledge may be relevant capabilities for the Swedish context and for inclusion in the CALY measure. This result informs the continued development of the CALY measure.

## Supporting information

**S1 Questionnaire. Survey questions in Swedish and translated to English.**
(DOCX)

## Acknowledgments

We thank the participants of this study.

## Author Contributions

**Conceptualization:** Anna Månsdotter, Björn Ekman, Inna Feldman, Lars Hagberg, Anna-Karin Hurtig, Lars Lindholm.

**Data curation:** Björn Ekman.

**Formal analysis:** Anna Månsdotter, Kaspar Walter Meili, Inna Feldman, Lars Hagberg.

**Funding acquisition:** Anna Månsdotter, Inna Feldman, Anna-Karin Hurtig, Lars Lindholm.

**Investigation:** Anna Månsdotter, Björn Ekman, Lars Hagberg.

**Methodology:** Kaspar Walter Meili, Inna Feldman, Lars Hagberg, Lars Lindholm.

**Project administration:** Anna Månsdotter, Lars Lindholm.

**Supervision:** Anna Månsdotter, Anna-Karin Hurtig, Lars Lindholm.

**Validation:** Björn Ekman, Kaspar Walter Meili.

**Visualization:** Kaspar Walter Meili.

**Writing – original draft:** Anna Månsdotter, Kaspar Walter Meili.

**Writing – review & editing:** Anna Månsdotter, Björn Ekman, Kaspar Walter Meili, Inna Feldman, Lars Hagberg, Anna-Karin Hurtig, Lars Lindholm.

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
