## [Decision Letter · Decision Letter 0]

9 Jul 2020

PONE-D-20-06320

Towards capability-adjusted life years in public health and social welfare: Results from a Swedish survey on ranking capabilities

PLOS ONE

Dear Dr. Meili,

Thank you for submitting your manuscript to PLOS ONE. After careful consideration, we feel that it has merit but does not fully meet PLOS ONE’s publication criteria as it currently stands. Therefore, we invite you to submit a revised version of the manuscript that addresses the points raised during the review process.

We look forward to receiving your revised manuscript.

Kind regards,

Jon Wardle

Academic Editor

PLOS ONE

Journal Requirements:

2. Please include additional information regarding the survey or questionnaire used in the study and ensure that you have provided sufficient details that others could replicate the analyses.

For instance, if you developed a questionnaire as part of this study and it is not under a copyright more restrictive than CC-BY, please include a copy, in both the original language and English, as Supporting Information. Moreover, please include more details on how the questionnaire was pre-tested, and whether it was validated.

3. In your Methods section, please provide additional information about the participant recruitment method and the demographic details of your participants.

Please ensure you have provided sufficient details to replicate the analyses such as:

a) the recruitment date range (month and year),

b) a description of any inclusion/exclusion criteria that were applied to participant recruitment,

c) a table of relevant demographic details.

Reviewers' comments:

Reviewer's Responses to Questions

**Comments to the Author**

1. Is the manuscript technically sound, and do the data support the conclusions?

Reviewer #1: Partly

Reviewer #2: Partly

Reviewer #3: Partly

2. Has the statistical analysis been performed appropriately and rigorously? 

Reviewer #1: Yes

Reviewer #2: Yes

Reviewer #3: Yes

3. Have the authors made all data underlying the findings in their manuscript fully available?

Reviewer #1: No

Reviewer #2: No

Reviewer #3: No

4. Is the manuscript presented in an intelligible fashion and written in standard English?

Reviewer #1: Yes

Reviewer #2: Yes

Reviewer #3: Yes

5. Review Comments to the Author

Reviewer #1: Thank you or the opportunity to review the paper, which aims to rank the importance of capabilities and to indicate relevant capabilities for the Swedish context. The paper is part of a larger project that aims to develop a measure for quality of life for policy making based on the CALYs framework. I think this is an important area of research but I believe the authors need to provide some further clarification and justification points for their chosen approaches, which are not only of importance for the current paper but the project overall. I have two major concerns related to the concept of CALYs as well as the methodology applied in this study.

Regarding the CALY concept, the authors criticize the QALY framework because it is limited to health. However, this is only partly correct. Since the QALY framework is based on the extra-welfarism approach, it can include also non-health benefits (see discussion by Brouwer et al. 2008). The problems is the operationalisation of the current QALY framework, where existing preference-based measure have a strong focus on health. However, this is not a limitation of the QALY framework but rather an issue with existing measures. Indeed, the authors suggest to develop a new measure that consist of 5 capability dimensions and 3 levels and will be valued using DCE and TTO, similar to the EQ-5D, which rather sounds that the authors propose a new preference-based measure, which could easily be implemented into the QALY framework. Secondly, the authors propose the collapsing of multidimensional capabilities information into a single index score using preference-based valuation techniques, which is in tension with Sen’s capability approach but not discussed by the authors. I guess these are comments related to the project overall but also relevant to understand the current paper.

My second major concern relates to the methods applied in this study. Firstly, it is generally known in the capability literature that the selection of capabilities needs to be context specific, and derived from a participatory approach. The authors state that the 10 capabilities were based on a review but it sounds like no interviews with members of the Swedish population have been undertaken. Secondly, the ranking exercise was conducted with researchers and doctoral students in health-related disciplines rather than member of the general population who may hold very different views. Finally, the authors conclude that the ranking results show that it is possible to discriminate between capabilities but fail to acknowledge the limitations of the ranking exercise. Previous literature discussed that ranking exercises do not involve trade-offs and, as such, respondents tend to ignore relative importance and say that everything is important, giving rise to ceiling effects.

Minor comments:

Introduction:

• On page 3, line 55, the authors mention valuable aspects that are currently omitted but do not discuss what these aspects are. Could the authors elaborate on that further?

• A further elaboration on QALYs and how they are derived is needed in the introduction.

• Line 76: could the authors discuss how Cookson’s interpretation of ‘capability QALYs’ is similar or differs to CALYs?

• The rationale for this study was not clear and how it will contribute to the overall project. Suggest re-phrasing the aims of this study in a more clear way.

Methods:

• Why did the authors choose researchers and doctoral students in health-related disciplines rather than member of the general population who may hold very different views?

• Why did the authors decide to use a ranking exercise over and above other techniques?

• The exclusion of participants due to reverse ranking of importance according to health is not clear and not enough justified. I think it is legitimate that some people may prefer other aspects more than health.

Discussion:

• The authors conclude that health was found to be the most important capability. However, I wonder to what extent this is driven by the exclusion of participants based on reverse ranking of importance according to health?

• It is not clear how the results of this study will be used. What is the contribution of this paper to the overall project, especially considering the fact that a valuation study (using DCE and TTO) will be used in future.

Reviewer #2: The study entitled “Towards capability-adjusted life years in public health and social welfare: Results from a Swedish survey on ranking capabilities” is relevant and provides an interesting concept to the reader. It addresses a noteworthy topic of capability-adjusted life years for supporting public health policies. Please find enclosed my comments.

Abstract:

The abstract reads fine, However, the authors mentioned that “overall purpose is to develop a measure for quality of life for policy making based on the capability approach by Amartya Sen”. The reader could be confused that the authors have developed the measure. If the authors would like to put developing the measure as their overall goal, they should at least mention the steps for doing so and the steps needed until they reach this purpose, however, that was out of the scope of this paper. I recommend to re-formulate this section, and to be more focused and clearer, e.g. the purpose is to provide the initial steps/evidence for developing a measure for quality of life for policy making based on the capability approach by Amartya Sen.

Introduction:

Please highlight the gap in literature and why this study is needed before the aim of the study by the end of introduction.

The Author could better tackle the rational of the study with using ranking to develop the measure, in other words, why ranking is important step for developing a measure for quality of life for policy making based on the capability approach.

In lines 55-56, the authors mentioned “Current approaches may omit valuable aspects and impacts and may fail to consider equity goals.” Please add the relevant references and clarify what are the current approaches accordingly.

Although the overall purpose was to contribute in developing a measure for quality of life, the authors didn´t have enough background about QoL and its main domains (e.g. according to the WHO) as well as and its measures. Also, QALYs and how it is linked with the capability approach. It is also important to add evidence about the use and how comparable or different CALYs and QALYs, that will clarify and strengthen the rational of the study. The authors have only pointed out very briefly about EQ-5D-3L.

Lines 119-120 “We envision a structure for CALY states that consists of 5 capability dimensions and 3 levels of achievement, totalling 243 possible states”. Its not clear how the 243 based on the 5 capabilities and 3 levels was reached. Please clarify.

From line 128 to 131 “ We found that ranking capabilities is feasible and that health, social relations, and financial situation were ranked highest by a non representative sample of researchers and doctoral students in health-related disciplines at five Swedish universities”, this paragraph should not be in the introduction, it is a finding so I recommend to move it to the discussion or results sections.

Methods:

Line 136, the author mentioned about piloting purposes, did the author do a pilot study or what did they mean in this section? a Pilot study is performed in a very small scale to be able to develop and improve the actual study on a full scale. Please clarify.

Regarding the study population what was the rational behind selecting the sample from health-related disciplines, and what does that means for the results?, because it is very important to reflect all views of capabilities in this ranking given that health related population might rank capabilities differently from the general population. Would the authors expect different results from different populations characteristics?

Please provide information about sampling technique (e.g. convenience, random,..). Did the authors calculate a sample size?

Statistical analysis: Line 203, “We calculated means and medians per strata” which strata?, in order to be clear, the authors can mention here the stratification done by gender, research field, and age group.

Results:

The number 115 was the sum of 112 who fulfilled the task correctly + 3 imputed responses. In this regard, line 215, it´s not clear where were the 28 respondents (17%) excluded from, and on which group did the authors perform their sensitivity analysis?

What is the rationale behind presenting both mean and median rank? Shouldn´t that be determined according to the data distribution and then choosing mean or median whichever would be meaningful for the data, what do the authors think about that?

In Table 1, some items are hidden visually within the table e.g. Political rights, please adjust.

In Table 1, what does it mean for the results that the ranked 1 to 5 column showing occupation less than time, security, and political rights, yet for the win count balance occupation was higher eventually?

Discussion:

Implication of this study and how it provides evidence and a step forward for developing CALYs needs to be emphasized.

How the study with its ranking strategy would contribute to the ultimate goal of developing a measure for quality of life based on capability approach, and what are the future needed steps?

Why the authors have focused on the top ranked 5 and not 6 or7?

Limitations:

The limitations mentioned by the authors were clear, however, lines from 388-392, didn´t seem like they belong in the limitations, e.g. “an info about exploring the calculation of CALY weights in a web-based survey among 2,000 Swedish residents”, this important statement should be discussed in the discussion instead including how that impact this study.

Overall, the study is well written and provide resources for the most important needed domains of capabilities, However, it needs to be very clear within the paper how this step contributes to the overall development of CALYs.

Reviewer #3: This paper reports results from a ranking exercise, undertaken by academics working in health related disciplines at five universities across Sweden. 10 broad reaching attributes (interpreted as capabilities) are identified from a Government review, for inclusion within the ranking exercise. There is an aspiration to eventually develop a five dimension measure, with a scoring system, which can be used to calculate capability-adjusted life years.

Whilst I support the rationale for developing such a measure, and whilst I agree that a case could be made for narrowing down the initial list of 10 capabilities to a manageable number through such a ranking exercise (with expert participants), I felt that there were a number of significant issues with the framing, methods, interpretation and contribution of the manuscript as it currently stands. Given the significance of these issues, I will not pick up on many specific points relating to presentation, but in relation to presentation, I will very briefly add that the message was unclear on page 4 (lines 85 to 89) and on page 5 (lines 115 to 118), and these sections will need to be rephrased to ensure clarity when resubmitting any future draft.

Framing and motivation:

- I felt that more information is needed about how the list of 10 capabilities were initially identified. I appreciate that this work was completed by external agencies and that a reference is given, but I had no idea why or how the 10 capabilities were identified or indeed how valid the process was. These 10 capabilities are the starting point/foundation for everything that the research team report here. Were they interpreted as capabilities by the agency that undertook work to identify them? On what basis did the research team come up with the short descriptions for the 10 attributes?

- What was the justification for (exclusively) recruiting health-related academics to complete the weighting task? Why not health/public health experts working in public sector/Government/local authority agencies? Why were health academics deemed to have an exclusively relevant perspective?

- Linked to the previous point, the researchers appear to have an a priori assumption that health should (perhaps even must) appear in the top 5, but little justification is given for this. I fear this a priori belief biases their perspective, with the suggestion that those academics not ranking health highly misinterpreted the scale. If there is a starting position that health should be in the top 5, why not simply include it

- why was it decided that 5 attributes were needed? This seems entirely arbitrary. The problem with this methodology is that it gives you no information on the relative VALUE of attributes and no information to decide whether 4 or 5 or 6 attributes stand out as being really important; so it relies on an arbitrary a priori decision.

Methods:

- It is not clear what background academics were given as to the nature of the task and why the ranking exercise was being undertaken. What perspective were academics asked to adopt (personal or societal)? Were they told what the short list of attributes would be used for? Were they expected to consider the policy agenda or things such as the current financial and environmental climate?

- This is a fundamentally basic task (OK, it requires people to make complex value judgements, but technically, a ranking task should be straightforward for an academic sample) and yet a third of respondents 'got it wrong'. I find this worrying and it needs further consideration and discussion.

Results:

- For such a simple task, the analysis is rather complicated! Numerous means of ranking attributes are reported but no indication is given as to which is most useful or most valid. It is seemingly entirely exploratory, but then there is seemingly also no conclusion / no recommendation from the work?

Discussion / conclusion:

- I am at a bit of a loss to see what the take home message of this paper is? The methods appear problematic, there is no clear recommendation of which five attributes should be taken forward to develop into a measure. It appears that other methods and other samples have been used to address the same question, but no indication of whether that gives us a definitive conclusion either. So I cannot see how any of the research results presented actually inform the overall research objective / future aspirations? Indeed the authors seem unsure themselves saying "This result may inform the continued development of the CALY measure".

6. PLOS authors have the option to publish the peer review history of their article (what does this mean?). If published, this will include your full peer review and any attached files.

Reviewer #1: No

Reviewer #2: **Yes: **Maisa Omara

Reviewer #3: No

---

## [Author Response · Author response to Decision Letter 0]

21 Sep 2020

Dear Reviewers and Editor

We would like to extend our gratitude and appreciation to you and the reviewers for your efforts in reviewing our manuscript and for providing important and helpful comments and suggestions. 

We incorporated the majority of the suggestions, and we think that the suggested changes substantially improved the quality of the manuscript. For example, the relation to the overall CALY project and future research is now described much clearer, and we clarified the limitations regarding that the participants have a background as researchers health-related fields. 

You find below with point-by-point responses to the individual comments. 

Our individual responses are listed below, grouped by reviewer and editor. 

Sometimes we refer to other comments using a numbering scheme: E stands for Editor, and R1, R2, or R2 for reviewer 1 – 3. The comment number refers to the number in the left column.

We also provide these comments in a more accessible form as a table in the seperate "Response to reviewers" file. 

Editor

1. Please ensure that your manuscript meets PLOS ONE's style requirements, including those for file naming. The PLOS ONE style templates can be found at …

Reply: Thank you for pointing out these shortcomings. We changed the filename of figure1.eps to Fig1.eps according to the style requirements. We added the city names to the affiliations. We also adjusted the font size of the headings and the intend of paragraphs to match the style templates.

2. Please include additional information regarding the survey or questionnaire used in the study and ensure that you have provided sufficient details that others could replicate the analyses.For instance, if you developed a questionnaire as part of this study and it is not under a copyright more restrictive than CC-BY, please include a copy, in both the original language and English, as Supporting Information.

Reply: Thank you for highlighting the need for providing the questionnaire. We added supporting information with questionnaire in both Swedish (original language) and English (for documentary purpose).

Where:

Supplementary information S1

2 (cont). Moreover, please include more details on how the questionnaire was pre-tested, and whether it was validated .

Reply: We added a sentence “We pre-tested the survey informally in our networks outside academia.” Under Procedure and questionnaire in the Methods section. We also added a sentence under Limitations: “Fourth, we did not a priori calculate a required sample size or validate the questionnaire.”

Where:

246-247, 461-462

3. In your Methods section, please provide additional information about the participant recruitment method and the demographic details of your participants.Please ensure you have provided sufficient details to replicate the analyses such as:a) the recruitment date range (month and year),

Reply: Thank you for suggesting adding additional information about the recruitment and participant characteristics. We agree that this information is crucial and needs to be presented clearly. The information on date range, recruitment, and survey process is provided under Procedure and questionnaire. To elaborate on the recruitment date range, we added a sentence: “The survey was accessible at once.” under Procedure and questionnaire. We also list the stratification criteria under Statistical analysis.

Where:

231, 257-258

3. b) a description of any inclusion/exclusion criteria that were applied to participant recruitment.

Reply: For clarification, we clarified that the study population was “Swedish-speaking” participants under the heading Study population. Beyond language, there were no inclusion/exclusion criteria besides the informed consent to participate.

Where:

208

3. c) table of relevant demographic details.

Reply: We added a new table (Table 1) with the number and proportion of participant background characteristics under Results. The numbering of following tables was increased by 1.

Where:

278

4. We note that you have indicated that data from this study are available upon request. PLOS only allows data to be available upon request if there are legal or ethical restrictions on sharing data publicly. For more information on unacceptable data access restrictions, please see http://journals.plos.org/plosone/s/data-availability#loc-unacceptable-data-access-restrictions.. a) If there are ethical or legal restrictions on sharing a de-identified data set, please explain them in detail (e.g., data contain potentially sensitive information, data are owned by a third-party organization, etc.) and who has imposed them (e.g., an ethics committee). Please also provide contact information for a data access committee, ethics committee, or other institutional body to which data requests may be sent.

Reply: We decided to publish a minimal anonymized dataset. Please see the cover letter for a detailed explanation.

Where:

Cover letter

b) Uploading the anonymized data set necessary to replicate the study findings.

Reply: See E4 a). We uploaded the data to the Swedish national data service. A DOI has been reserved but is not yet accessible due to pending review. In the meantime, we provide the dataset as part of the submission.

Where:

HYPERLINK "https://doi.org/10.5878/r2nm-zc35" https://doi.org/10.5878/r2nm-zc35

Reviewer #1

1st main. Regarding the CALY concept, the authors criticize the QALY framework because it is limited to health. However, this is only partly correct. Since the QALY framework is based on the extra-welfarism approach, it can include also non-health benefits (see discussion by Brouwer et al. 2008). The problems is the operationalisation of the current QALY framework, where existing preference-based measure have a strong focus on health. However, this is not a limitation of the QALY framework but rather an issue with existing measures. Indeed, the authors suggest to develop a new measure that consist of 5 capability dimensions and 3 levels and will be valued using DCE and TTO, similar to the EQ-5D, which rather sounds that the authors propose a new preference-based measure, which could easily be implemented into the QALY framework.

Reply: Thank you for pointing out this important issue. We agree that this needs to be clarified and our intent was to convey that QALY is of extra-welfarist nature and not limited to health. We added a sentence “Due to the extra-welfarist nature, the QALY concept is in theory not restricted to measuring health benefits but in practice mostly used to measure health consequences“ under the heading Theoretical issues.

Where:

94-95

2nd main. Secondly, the authors propose the collapsing of multidimensional capabilities information into a single index score using preference-based valuation techniques, which is in tension with Sen’s capability approach but not discussed by the authors. I guess these are comments related to the project overall but also relevant to understand the current paper.

Reply: Thank you for your comment and helpful suggestion. We agree that the capability approach as outlined by Sen conflicts in this aspect with the proposed CALY measure and that it is crucial to discuss. We discuss this now under Other developments and our proposal

Where:

156-161

3rd main. My second major concern relates to the methods applied in this study. Firstly, it is generally known in the capability literature that the selection of capabilities needs to be context specific, and derived from a participatory approach. The authors state that the 10 capabilities were based on a review but it sounds like no interviews with members of the Swedish population have been undertaken.

Reply: Thank you for pointing out this issue. We agree with your concern. The aim was to investigate the ranking of capabilities in a survey. The context specificity is given by using a Swedish example. We have clarified that a participatory Delphi process approach using the results from this ranking study was planned and has now been conducted (manuscript in preparation), under Future research. We also provide more context about the governmental investigation that selected the 10 capabilities under Material and Methods.

Where:

466-473, 175-183

4th main. Secondly, the ranking exercise was conducted with researchers and doctoral students in health-related disciplines rather than member of the general population who may hold very different views.

Reply: Thank you for sharing your concern. We acknowledge this limitation. The aim was to investigate the feasibility of ranking and indicate a tentative selection. We clarified the limitation in the Limitations section with a sentence: “The study participants were not representative of the Swedish population and consisted of researchers in health-related disciplines. Hence, they are likely biased towards valuing health, but also other capabilities such as knowledge and social relations as determinants of health, as particularly important. “ We also provide more detail on the professional background under Study population, the first sentence now reads “The study population consisted of Swedish-speaking researchers and doctoral students in the disciplines of public health, global health, epidemiology, medicine, women´s and children´s health, and health care at five Swedish universities.”

Where:

445-450, 208-212

5th main. Finally, the authors conclude that the ranking results show that it is possible to discriminate between capabilities but fail to acknowledge the limitations of the ranking exercise. Previous literature discussed that ranking exercises do not involve trade-offs and, as such, respondents tend to ignore relative importance and say that everything is important, giving rise to ceiling effects.

Reply: We agree that it is important to avoid ceiling effects and thank you for considering this important issue. To promote trade-offs, we chose to use a ranking task where participants were instructed to assign unique ranks from 1 to 10 to the dimensions, as opposed to for example rating where respondents would assign values from 1 (least) to 10 (most important) to rate their importance. This could result in valuing all dimensions as 10. We clarified this property under the Methods heading by inserting the sentence “The intent of using a ranking task was to promote trade-offs between the capabilities and avoid participants indicating all capabilities as important.”

Where:

239-241

Introduction

1st minor. On page 3, line 55, the authors mention valuable aspects that are currently omitted but do not discuss what these aspects are. Could the authors elaborate on that further? 1st minor

Reply: Thank you for your suggestion. We reworded and extended the on the wording to clarify the context.

Where:

64-69

2nd minor. A further elaboration on QALYs and how they are derived is needed in the introduction.

Reply: Thank you for your suggestion, we agree and included a description of QALYS and how QALY weights are derived.

Where:

84-92

3rd minor. could the authors discuss how Cookson’s interpretation of ‘capability QALYs’ is similar or differs to CALYs?

Reply: Thank you for your comment. We changed and expanded the wording to clarify: “Due to the extra-welfarist nature, the QALY concept is in theory not restricted to measuring health benefits, but in practice is used mostly to evaluate health consequences. Cookson [16] suggested a ‘capability QALY’ interpretation of QALYs as a “cardinal index of an individual’s capability set”. We see CALYs, as outlined below, as a specific implementation according to Cookson’s interpretation.”

Where:

94-98

4th minor. The rationale for this study was not clear and how it will contribute to the overall project. Suggest re-phrasing the aims of this study in a more clear way.

Reply: We inserted a sentence after the aim: “In relation to the overall project, the purpose was to collect information for the selection of dimensions for the CALY measure.” See also 9th minor.

Where:

168-170

Methods

5th minor. Why did the authors choose researchers and doctoral students in health-related disciplines rather than member of the general population who may hold very different views?

Reply: Thank you for your comment, please see R1, 4th main.

6th minor. Why did the authors decide to use a ranking exercise over and above other techniques?

Reply: Thank you for your comment, please see R1, 5th main.

7th minor. The exclusion of participants due to reverse ranking of importance according to health is not clear and not enough justified. I think it is legitimate that some people may prefer other aspects more than health.

Reply: Thank you for your suggestion. We further clarified how the sensitivity analysis was done under Methods and Results->Sensitivity analysis. See also R2, 14, and R3, 3.

Where:

248-254, 354-368

Discussion

8th minor. The authors conclude that health was found to be the most important capability. However, I wonder to what extent this is driven by the exclusion of participants based on reverse ranking of importance according to health?

Reply: Thank you for your inquiry. The exclusion based on suspected reverse rankings was only done for the sensitivity analysis. The main results, which the conclusions are based on, did not exclude participants. Furthermore, the ranking of dimensions was similar in both the main and the sensitivity analysis in all the metrics. See also R1, 7th minor.

9th minor. It is not clear how the results of this study will be used. What is the contribution of this paper to the overall project, especially considering the fact that a valuation study (using DCE and TTO) will be used in future.

Reply: Thank you for highlighting this shortcoming. We clarified the next steps of the development in the section Future research, specifically how the results were used to conduct a Delphi process. We further deleted “may” from the concluding sentence which now reads: “This result informs the continued development of the CALY measure.” Please also see R1, 4th minor.

Where:

481, 472-476

Reviewer #2

Abstract

1. The abstract reads fine, However, the authors mentioned that “overall purpose is to develop a measure for quality of life for policy making based on the capability approach by Amartya Sen”. The reader could be confused that the authors have developed the measure. If the authors would like to put developing the measure as their overall goal, they should at least mention the steps for doing so and the steps needed until they reach this purpose, however, that was out of the scope of this paper. I recommend to re-formulate this section, and to be more focused and clearer, e.g. the purpose is to provide the initial steps/evidence for developing a measure for quality of life for policy making based on the capability approach by Amartya Sen.

Reply: Thank you for this important observation and helpful suggestions. We agree that it is confusing. We reworded the abstract introduction to “In this study, the aim was to rank the importance of capabilities and to indicate relevant capabilities for the Swedish context to inform the development of capability-adjusted life years (CALYs), a quality of life measure for policy making based on the capability approach by Amartya Sen.” See also R1, 9th minor

Where:

29-32

Introduction

2. Please highlight the gap in literature and why this study is needed before the aim of the study by the end of introduction.

Reply: Thank you for your suggestion. We inserted “To our knowledge, no previous work specific to the Swedish context exists that has attempted to assess a relevant selection of capabilities based on survey data.” before the aim.

Where:

163-165

3. The Author could better tackle the rational of the study with using ranking to develop the measure, in other words, why ranking is important step for developing a measure for quality of life for policy making based on the capability approach.

Reply: We agree that the rationale in relation to the whole project was unclear, thank you for highlighting this. We clarified the rationale behind a ranking task, please see R1, 5th major, as well as the implications for developing the CALY measuring, please see R1, 9th minor.

4. In lines 55-56, the authors mentioned “Current approaches may omit valuable aspects and impacts and may fail to consider equity goals.” Please add the relevant references and clarify what are the current approaches accordingly.

Reply: Thank you for your comment, we agree that this is an unclearly formulated. Please see R1, 1st minor

5. Although the overall purpose was to contribute in developing a measure for quality of life, the authors didn´t have enough background about QoL and its main domains (e.g. according to the WHO) as well as and its measures.

Reply: Thank you for your suggestion. We agree that it would be informative to have a comprehensive background section on these topics. However, the relevant literature is vast and, in our opinion, goes beyond the scope of this paper. We attempted to find a way of including the WHO definition of quality of life, but we did not succeed in doing so without inadequately prolonging the text. Instead, we suggest to keep to introducing the topic from a (health) economic perspective and the according historical context involving welfarism and extra-welfarism to give the reader an appropriate background.

6. Also, QALYs and how it is linked with the capability approach.

Reply: Thank you for your comment. We provide a description of extra-welfarism and capabilities, and how the capability approach may be implemented in a QALY-like manner under the headings Theoretical issues and Other approaches and our proposal. For example, we reiterate Cookson’s suggestion of the capability QALY. We clarified this relation further. Please also see R1, 3rd minor.

Where:

84-98, 139-146

7. It is also important to add evidence about the use and how comparable or different CALYs and QALYs, that will clarify and strengthen the rational of the study.

Reply: Thank you for emphasizing the need to clearly differentiate CALY and QALYs. We inserted a sentence “Whereas QALYs are appropriate in situations where the main interest are health benefits, CALYs aim to be used in situations where the goal is to measure changes in individuals’ general capability sets.” Under the heading Other developments and our proposal.

Where:

144-146

8. The authors have only pointed out very briefly about EQ-5D-3L.Lines 119-120 “We envision a structure for CALY states that consists of 5 capability dimensions and 3 levels of achievement, totalling 243 possible states”. Its not clear how the 243 based on the 5 capabilities and 3 levels was reached. Please clarify.

Reply: Thank you for y pointing out this shortcoming. We describe EQ-5D now more detailed as an example for QALY, see R1, 2nd minor. The number of different states, or profiles, for an instrument with 5 levels with each 3 dimensions is 243, because 3^5 =243. We altered this section to be clearer, please see R3, 4.

9. From line 128 to 131 “ We found that ranking capabilities is feasible and that health, social relations, and financial situation were ranked highest by a non representative sample of researchers and doctoral students in health-related disciplines at five Swedish universities”, this paragraph should not be in the introduction, it is a finding so I recommend to move it to the discussion or results sections.

Reply: Thank you for your comment. We agree that normally statements mentioning results should not be in the introduction. However, we placed this sentence according to the PLOSOne submission guidelines: “Conclude [the introduction] with a brief statement of the overall aim of the work and a comment about whether that aim was achieved” (https://journals.plos.org/plosone/s/submission-guidelines#loc-supporting-information). We asked the editor for clarification regarding this issue.

Where:

Cover letter

Methods

10. Line 136, the author mentioned about piloting purposes, did the author do a pilot study or what did they mean in this section? a Pilot study is performed in a very small scale to be able to develop and improve the actual study on a full scale. Please clarify.

Reply: Thank you for pointing out this wording. We removed “piloting”. We agree that the wording is misleading here.

Where:

181

11. Regarding the study population what was the rational behind selecting the sample from health-related disciplines, and what does that means for the results?, because it is very important to reflect all views of capabilities in this ranking given that health related population might rank capabilities differently from the general population. Would the authors expect different results from different populations characteristics?

Reply: Thank you for comment and valid concerns. Please see see R1, 4th main.

12. Please provide information about sampling technique (e.g. convenience, random,...). Did the authors calculate a sample size?

Reply: We clarified under Study population that it was a convenience sample. We did not calculate a sample size requirement. We added an additional sentence under Limitations that reads “Fourth, we did not a priori calculate a required sample size and validate the questionnaire.” to clarify this.

Where:

208-209, 461-462

13. Statistical analysis: Line 203, “We calculated means and medians per strata” which strata?, in order to be clear, the authors can mention here the stratification done by gender, research field, and age group.

Reply: Thank you for you proposition, we added the list of strata as you suggested, the sentence reads now: “We calculated means and medians overall and stratified by gender, research field, and age group.”

Where:

257-258

Results

14. The number 115 was the sum of 112 who fulfilled the task correctly + 3 imputed responses. In this regard, line 215, it´s not clear where were the 28 respondents (17%) excluded from, and on which group did the authors perform their sensitivity analysis?

Reply: Thank you for your comment. We group the sensitivity analysis results under a new subheading under results, Sensitivity analysis. To make it clear where the respondents were excluded from, we used the sentence: “For the sensitivity analysis, in which these 28 potential reverse rankings according to the health criterion were excluded from the 115 respondents in the main analysis, the rankings of importance varied little.”

Where:

358-360

15. What is the rationale behind presenting both mean and median rank? Shouldn´t that be determined according to the data distribution and then choosing mean or median whichever would be meaningful for the data, what do the authors think about that?

Reply: Thank you for your comment. We agree that appropriate metrics should be chosen based on distribution characteristics of data. In addition, the nature of the scale, in our case, ordinal ranks, is of importance. We present the distributions of ranks by dimension in Figure 1. The shape of the distribution varies, and skewed distributions may indicate that the mean is not appropriate.However, we primarily chose to report multiple metrics because they each have their advantages and disadvantages, as discussed in the section “Ranking of importance and comparison with other studies”. We clarified this section further. Specifically, means for ordinal data are generally less because they imply that the difference between ranks indicates their relative strength to each other. The median however is the middle of the distributions which is more appropriate for ranks. If occupation ranks 1,1, and 10, the median would be 1 whereas the mean would be 4. Other persons may value occupation the same but other dimensions differently, resulting in the ranks of 1,1, and 4 with mean of 2 and median of 1. The median results in the same rank, but the mean is influenced by how the other dimensions are valued and becomes different, even though both value occupation the same. However, the median rank for two dimensions may be the same even though their distribution varies, for example for 1, 2,2 and 2, 2, 3. In that case, they mean may help to resolve a tie. For a light-read discussion see for example here: https://measuringu.com/mean-ordinal/

16. In Table 1, some items are hidden visually within the table e.g. Political rights, please adjust .

Reply: Thank you for pointing this out. We corrected the issue.

Where:

305

17. In Table 1, what does it mean for the results that the ranked 1 to 5 column showing occupation less than time, security, and political rights, yet for the win count balance occupation was higher eventually?

Reply: Thank you for your question. It means that occupation was more often ranked ahead of Time, security, political rights, housing, and living environment, but less often in the top five than time, security, and political rights. Consider for example ranks 7, 8; 8, 9; 8, 2 for occupation and time respectively. Occupation wins 2 times against time and loses once, resulting in a win count balance of 2 out of 3 rankings. Yet it is never ranked in the top 5 whereas time is once. The win count balance says something about how many times a dimension was ranked ahead of another one, whereas the top 5 count is more of an overall assessment across all dimensions. We added a sentence “For example, a win count balance of 5 for A over B means that A was ranked 5 times ahead of B.” to better illustrate the concept under the heading Statistical analysis.

Where:

263-265

Discussion

18. Implication of this study and how it provides evidence and a step forward for developing CALYs needs to be emphasized.How the study with its ranking strategy would contribute to the ultimate goal of developing a measure for quality of life based on capability approach, and what are the future needed steps?

Reply: Thank you for your comment, we agree that the overall picture needs to be clearer. We clarified this under Future research. Please see also R1, 4th minor.

Where:

472-476

19. Why the authors have focused on the top ranked 5 and not 6 or7?

Reply: Thank you for highlighting this. Our intent was to explicitly focus on the 5 highest ranked capabilities but rather to arrive at a tentative selection of most relevant capabilities for the Swedish context. Please also see R3, 4.

20. The limitations mentioned by the authors were clear, however, lines from 388-392, didn´t seem like they belong in the limitations, e.g. “an info about exploring the calculation of CALY weights in a web-based survey among 2,000 Swedish residents”, this important statement should be discussed in the discussion instead including how that impact this study.

Reply: Thank you for pointing out these unclarities. We agree that the limitations should be clearly distinguishable from future research. We split the Limitations and future research section into two headings, Limitations and Future research to make the distinction clearer and reported the information on the pilot study under the future research section. We clarified the wording, under Future research, because the wording “selecting” is confusing as no selection was done in the study; instead respondents traded-off between different capability states. A direct comparison with the ranking exercise is thus less appropriate and we prefer to keep it in the future research section where this information is strictly related to the planned next steps.

Where:

444, 463, 470-472

21. Overall, the study is well written and provide resources for the most important needed domains of capabilities, However, it needs to be very clear within the paper how this step contributes to the overall development of CALYs.

Reply: Thank you very much for your assessment. We agree that the relation the overall project needs to be communicated more clearly. Please see R2, 20. See R1, 4th and 9th minor.

Reviewer #3

0. … in relation to presentation, I will very briefly add that the message was unclear on page 4 (lines 85 to 89) and on page 5 (lines 115 to 118), and these sections will need to be rephrased to ensure clarity when resubmitting any future draft.

Reply: Thank you for pointing out these unclear formulations. We clarified these sections: “In democratic high-income countries such as Sweden, one category comprises capabilities that everybody has an undisputed right to have access to, such as political resources or a minimal threshold of education through basic schooling. For other capabilities, there may be less consensus that everybody should have them to the same extent, for example, the capability to be wealthy enough to buy property or to have an occupation that one is always completely happy with. “ And “For example, a state that represents the quality of life for a given capability set may have a weight of 0.7 attached, resulting in 0.7 CALYs for a person-year, i.e., the same concept as in QALYs.”

Where:

108-113, 140-144

Framing and motivation

1. I felt that more information is needed about how the list of 10 capabilities were initially identified. I appreciate that this work was completed by external agencies and that a reference is given, but I had no idea why or how the 10 capabilities were identified or indeed how valid the process was. These 10 capabilities are the starting point/foundation for everything that the research team report here. Were they interpreted as capabilities by the agency that undertook work to identify them? On what basis did the research team come up with the short descriptions for the 10 attributes?

Reply: Thank you for pointing out the important necessity to discuss the origin of the 10 capabilities in more detail. We added details the governmental investigation, on the compilation of the list of 10 capabilities and how their descriptions were derived in the manuscript under List of suggested capabilities.

Where:

175-183

2. What was the justification for (exclusively) recruiting health-related academics to complete the weighting task? Why not health/public health experts working in public sector/Government/local authority agencies? Why were health academics deemed to have an exclusively relevant perspective?

Reply: Thank you for your comment. We acknowledge the limitation of this convenience sampling and clarified under Limitations. Please also see R1, 4th main.

Where:

446-450

3. Linked to the previous point, the researchers appear to have an a priori assumption that health should (perhaps even must) appear in the top 5, but little justification is given for this. I fear this a priori belief biases their perspective, with the suggestion that those academics not ranking health highly misinterpreted the scale. If there is a starting position that health should be in the top 5, why not simply include it

Reply: Thank you for your comment. This is a fair limitation, given we and the respondents come from health-related disciplines ourselves. We did not consider including health from the beginning as we do not want to take a prior decision but aim to conduct a Delphi process with “fair-minded” participants, that relies among other on the findings from this study. Additionally, the respondents may be highly aware of the so-called social determinants of health, i.e. that the other potential capabilities may be essential factors in itself. Please see also R1, 4th main. We did not just suspect potential reverse rankings based on the rank of health, but because often health was ranked 6-10 together with dimensions that were otherwise also commonly ranked 1-5, specifically social relations and financial situation. We added now more details on that and grouped results related to the sensitivity analysis under a new heading Sensitivity analysis, as well as under Procedure and questionnaire in the methods section.

Where:

248-254, 355-360

4. why was it decided that 5 attributes were needed? This seems entirely arbitrary. The problem with this methodology is that it gives you no information on the relative VALUE of attributes and no information to decide whether 4 or 5 or 6 attributes stand out as being really important; so it relies on an arbitrary a priori decision.

Reply: Thank you for your comment. We agree that the decision for 5th attributes is arbitrary and that no direct information about relative value of the attributes is provided by the ranking task. However, indirectly the variance among the mid ranks suggests that the dimensions ranked in the mid-tier are of similar value. Health, financial situation, and social relation however are clearly preferred. To clarify, under Other developments and our proposal, we removed the wording “envision a configuration of 5 dimensions with 3 levels”. We inserted the following sentence: “We envision a simple structure for CALY states with a limited number of capability dimensions and levels of achievement, similar to the EQ-5D-3L construction [28], for the purpose of facilitating both the further development and application of CALYs. The definite configurations of dimensions and levels still needs to be determined”. We additionally added a limitation under Limitations to clarify this limitation of the ranking exercise: “Deciding on how many capabilities should be included in the CALY measure based on ranking data may be arbitrary because the ranking task does not directly yield information about the relative strengths of the different capabilities.”

Where:

151-155, 451-453

Methods

5. It is not clear what background academics were given as to the nature of the task and why the ranking exercise was being undertaken . What perspective were academics asked to adopt (personal or societal )? Were they told what the short list of attributes would be used for? Were they expected to consider the policy agenda or things such as the current financial and environmental climate?

Reply: Thank you for bringing up this point. We describe the survey procedure, including the provided information, under the heading Procedure and questionnaire: “The attached documents consisted of an informational letter (background, aim, task, estimated time taken, risks and benefits, handling of data, presentation of results, and voluntariness) and a summarised research plan (the overall project and the study). This was followed by a reminder (10 June, 2016).”We also added a supporting information file with the exact question phrasings, please see E2.

Where:

Supplementary information S1

6. This is a fundamentally basic task (OK, it requires people to make complex value judgements, but technically, a ranking task should be straightforward for an academic sample) and yet a third of respondents 'got it wrong'. I find this worrying and it needs further consideration and discussion.

Reply: Thank you for your comment. We agree that this limitation needs to be clarified. We reworded and added details on this issue, the text describing it under Limitations now says: “Similarly, the occurrence of suspected inverse rankings is an indicator that the instructions and visual design of the survey need improvement.”

Where:

459-461

Results

7. For such a simple task, the analysis is rather complicated! Numerous means of ranking attributes are reported but no indication is given as to which is most useful or most valid. It is seemingly entirely exploratory, but then there is seemingly also no conclusion / no recommendation from the work?

Reply: Thank you for pointing out this important issue. We think that there is not a “one size fits” all metric for this problem hence the presentation of multiple metrics, as they all have their advantages and disadvantages which we discuss under Ranking of importance and comparison with other studies. Please also see also R2, 15. We present the clarified conclusions under Conclusion and partially under Future research.

Where:

404-414

Discussion / conclusion:

8. I am at a bit of a loss to see what the take home message of this paper is? The methods appear problematic, there is no clear recommendation of which five attributes should be taken forward to develop into a measure. It appears that other methods and other samples have been used to address the same question, but no indication of whether that gives us a definitive conclusion either. So I cannot see how any of the research results presented actually inform the overall research objective / future aspirations? Indeed the authors seem unsure themselves saying "This result may inform the continued development of the CALY measure".

Reply: Thank you for highlighting the unclarities regarding the take home message and the relation to the overall project. We clarified the conclusions to explicitly state the relation to future research: “The analysis shows that health, social relations, financial situation, and potentially knowledge may be relevant capabilities for the Swedish context. and for inclusion in the CALY measure. This result informs the continued development of the CALYs measure.’ We also provide more details under Future research how the overall research project profits from the findings. Please see also R1, 4th minor for more details on these clarifications.

Where:

478-482

---

## [Decision Letter · Decision Letter 1]

9 Nov 2020

Towards capability-adjusted life years in public health and social welfare: Results from a Swedish survey on ranking capabilities

PONE-D-20-06320R1

Dear Dr. Meili,

We’re pleased to inform you that your manuscript has been judged scientifically suitable for publication and will be formally accepted for publication once it meets all outstanding technical requirements.

Kind regards,

Paul Anand

Academic Editor

PLOS ONE

Additional Editor Comments (optional):

Reviewers' comments:

Reviewer's Responses to Questions

**Comments to the Author**

1. If the authors have adequately addressed your comments raised in a previous round of review and you feel that this manuscript is now acceptable for publication, you may indicate that here to bypass the “Comments to the Author” section, enter your conflict of interest statement in the “Confidential to Editor” section, and submit your "Accept" recommendation.

Reviewer #1: All comments have been addressed

2. Is the manuscript technically sound, and do the data support the conclusions?

Reviewer #1: Yes

3. Has the statistical analysis been performed appropriately and rigorously? 

Reviewer #1: Yes

4. Have the authors made all data underlying the findings in their manuscript fully available?

Reviewer #1: Yes

5. Is the manuscript presented in an intelligible fashion and written in standard English?

Reviewer #1: Yes

6. Review Comments to the Author

Reviewer #1: The authors have addressed all my comments as well as the comments from other reviewers. The clarity of the paper has improved and the authors have acknowledged important limitations of their study.

7. PLOS authors have the option to publish the peer review history of their article (what does this mean?). If published, this will include your full peer review and any attached files.

Reviewer #1: No

---

## [Editor Report · Acceptance letter]

18 Nov 2020

PONE-D-20-06320R1 

Towards capability-adjusted life years in public health and social welfare: Results from a Swedish survey on ranking capabilities 

Dear Dr. Meili:

I'm pleased to inform you that your manuscript has been deemed suitable for publication in PLOS ONE. Congratulations! Your manuscript is now with our production department. 

Kind regards, 

on behalf of

Professor Paul Anand 

Academic Editor

PLOS ONE